# Hyperpolarized water through dissolution dynamic nuclear polarization with UV-generated radicals

Arthur C. Pinon[1,2], Andrea Capozzi[1,2] & Jan Henrik Ardenkjær-Larsen [1✉]

In recent years, hyperpolarization of water protons via dissolution Dynamic Nuclear Polarization (dDNP) has attracted increasing interest in the magnetic resonance community. Hyperpolarized water may provide an alternative to Gd-based contrast agents for angiographic and perfusion Magnetic Resonance Imaging (MRI) examinations, and it may report on chemical and biochemical reactions and proton exchange while performing Nuclear Magnetic Resonance (NMR) investigations. However, hyperpolarizing water protons is challenging. The main reason is the presence of radicals, required to create the hyperpolarized nuclear spin state. Indeed, the radicals will also be the main source of relaxation during the dissolution and transfer to the NMR or MRI system. In this work, we report water magnetizations otherwise requiring a field of 10,000 T at room temperature on a sample of pure water, by employing dDNP via UV-generated, labile radicals. We demonstrate the potential of our methodology by acquiring a $^{15}N$ spectrum from natural abundance urea with a single scan, after spontaneous magnetization transfer from water protons to nitrogen nuclei.

[1] Center for Hyperpolarization in Magnetic Resonance, Department of Health Technology, Technical University of Denmark, Building 349, 2800 Kgs Lyngby, Denmark. [3]These authors contributed equally: Arthur C. Pinon, Andrea Capozzi. ✉email: jhar@dtu.dk

Since its birth, water protons have played a crucial role for Nuclear Magnetic Resonance (NMR). The detection of their nuclear spin magnetism has been employed to study relaxation phenomena[1], chemical exchange[2], material absorption properties[3], transport processes[4], just to name a few. Also, Magnetic Resonance Imaging (MRI) exploits the magnetism from water protons of the human body to non-invasively visualize organs and tissues[5]. Nevertheless, all these applications suffer from a main drawback: low sensitivity[6]. The reason is the weak nuclear magnetic moment of [1]H nuclei that leads to only few tens of ppm net polarization of the spins at ordinary values of magnetic field and room temperature. Over the years, the general approach to this issue has been to develop higher and higher magnetic field strengths. However, this comes at an exorbitant cost.

Hyperpolarization via dissolution Dynamic Nuclear Polarization (dDNP) was introduced in 2003 by Ardenkjær-Larsen and co-workers[7], and it has become the most widespread and versatile hyperpolarization method to overcome the low sensitivity of NMR in the liquid state. The enhancement of the NMR signal relies on the microwave driven polarization transfer from dilute unpaired electron spins to the surrounding nuclear spins at low temperature (1–4 K) and moderate magnetic field strength (3–7 T), followed by a fast dissolution of the sample. Sensitivity improvement, up to 4 orders of magnitude, gifted NMR with unprecedented temporal resolution and paved the way to new applications, such as fast chemical reaction monitoring[8,9], observation of protein folding in real time[10] and cancer diagnosis/ response to treatments in humans[11,12].

Although the technique has mainly been used for [13]C hyper-polarized MR spectroscopy and imaging, most recently, hyper-polarization of water protons has attracted increasing interest in the MR community. Indeed, hyperpolarized (HP) water has already been demonstrated to obtain high contrast angiographic and perfusion images in small and medium size animal models with no need for any paramagnetic metallic compound (e.g., $Gd^{3+}$)[13,14], or non-standard MRI equipment and sequences as for HP [13]C experiments[15]. Moreover, fast exchange with HP water [1]H nuclei has been used to enhance the sensitivity and reduce the scanning time in 1D and 2D MRS experiments on biomolecules dynamics, protein structure determination and protein–ligands interaction[16–20]. These studies have demonstrated the potential of HP water as an eclectic analytical tool; however, its use is still limited since hyperpolarization of water protons is challenging. In dDNP, the hyperpolarization is generated ex situ in the so-called polarizer. Therefore, a "sine qua non" condition for hyperpolarization is a relatively long nuclear spin relaxation time during the transfer to the measuring apparatus. Water protons have been efficiently polarized (30–40%) in the solid state already at traditional dDNP conditions (3.35 T and 1.2 K)[21], and close to unity polarization has been achieved by doubling the magnetic field[22,23]. Nevertheless, preserving this high spin order in the liquid-state is far from trivial. Water as such, characterized by a $T_1$ of approx. 3.5 s[24], is not a molecule suitable for dDNP. Compared to [13]C, the large magnetic moment and density of protons guarantee high and fast DNP in the solid state, when broad ESR line radicals such as nitroxides are used[25]. However, the water protons are exposed to severe relaxation due to strong dipolar couplings between [1]H nuclei themselves as well with the unpaired electron spin of the radicals. Therefore, prior to any application, it is imperative to prolong the $T_1$ of the water protons. This involves four main cruxes[13]. Firstly, molecular oxygen is paramagnetic and has to be removed from the sample. Secondly, dissolving the sample in $D_2O$ reduces the proton concentration. Thirdly, keeping the HP final solution at an elevated temperature increases the $T_1$ significantly[13]. Fourthly, the

radical used in the DNP process has to be eliminated. While the first three points are optimized in a straightforward manner[13,23], efficient and fast removal of the radical remains an open challenge. So far, three approaches have been pursued: scavenging of the radicals by ascorbic acid[13,26], extraction of non-polar radicals into an organic phase immediately after dissolutions[17,19,23], and filtration of the radicals covalently bonded to polymer-based backbones[27]. All these methods suffer from a common drawback: the process is not instantaneous and cause relaxation during the dissolution. From sample melting and dilution to radical removal, the water protons relax fast because of the strong dipolar coupling to the electron spins. As a consequence, although $T_1$ values >30 s have been recorded by matching the three conditions mentioned above, water proton polarization never exceeded 13.0 % in the final solution, implying a polarization loss during dissolution between 5 and 10 times depending on the experimental setup[13,19,23,27]. It is worth mentioning that other techniques such as Parahydrogen Induced Polarization (PHIP) and Overhauser DNP are suitable to generate HP water[28,29]. Nevertheless, the small enhancement and/or short relaxation time that characterize these alternative methods represent a main drawback when it comes to applications.

UV-induced labile radicals have been employed to efficiently hyperpolarize [13]C and other low-gamma nuclei by dDNP[30–34]. UV-irradiation of a frozen solution containing a fraction of pyruvic acid (PYR) or PYR derivatives generates radicals that are stable below 190 K[35]. These radicals recombine into diamagnetic species during the dissolution.

Taking advantage of this property and optimizing the radical precursor, here we establish a robust method to efficiently hyperpolarize water protons in the solid state and minimize polarization losses during and after dissolution.

## Results
**Apparatus**. Figure 1 shows a sketch of the instrumentation used in this study. The dDNP polarizer, operating at 6.7 T and 1.15 ± 0.05 K, is shown on the left-hand side of panel **a**. It is conceptually similar to the idea introduced in 2003[7], but the sample insertion unit was modified to accommodate a custom fluid path (CFP) dissolution system. The CFP is an evolution of the previously described fluid path (see Fig. 1, panel **e**)[32,36]. The new version not only is reusable and suitable for the loading of frozen solid samples, but also employs helium chase gas to expel the sample (for more details see Methods). The polarizer and a 9.4 T NMR magnet (right hand side of Fig. 1, panel **a**) are connected via a 2.6 m long magnetic tunnel[37]. The two magnets both have the north pole in the same direction. The tunnel provides a homogeneous vertical field of 0.55 T across the full length of the transfer line connecting the outlet of the CFP (polarizer side) and the inlet of the NMR tube (NMR magnet side). The magnetic field experienced by the HP solution during transfer with and without the magnetic tunnel was measured using a Hall probe and shown in panel **b** and **c**, respectively. A schematic representation of the magnetic tunnel profile is shown in panel **d** (for more details see Methods).

**Sample formulation and UV-radical generation**. Two UV-radical precursors were considered in this study: natural abundance pyruvic acid (PYR) and [2-$^{13}$C]pyruvic acid (2CPYR). Three mixtures were prepared to generate UV-irradiated dDNP samples: PYR:glycerol-$d_8$:$H_2O$ 2:3:5 (v/v/v); 2CPYR:glycerol-$d_8$:$H_2O$ 2:3:5 (v/v/v) and 2CPYR:glycerol-$d_8$:$D_2O$:$H_2O$ 2:3:4:1 (v/v/v/v). The three preparations are referenced as PYR_sample, 2CPYR_sample and 2CPYRd_sample, respectively. The formulations were chosen to study the influence of two parameters

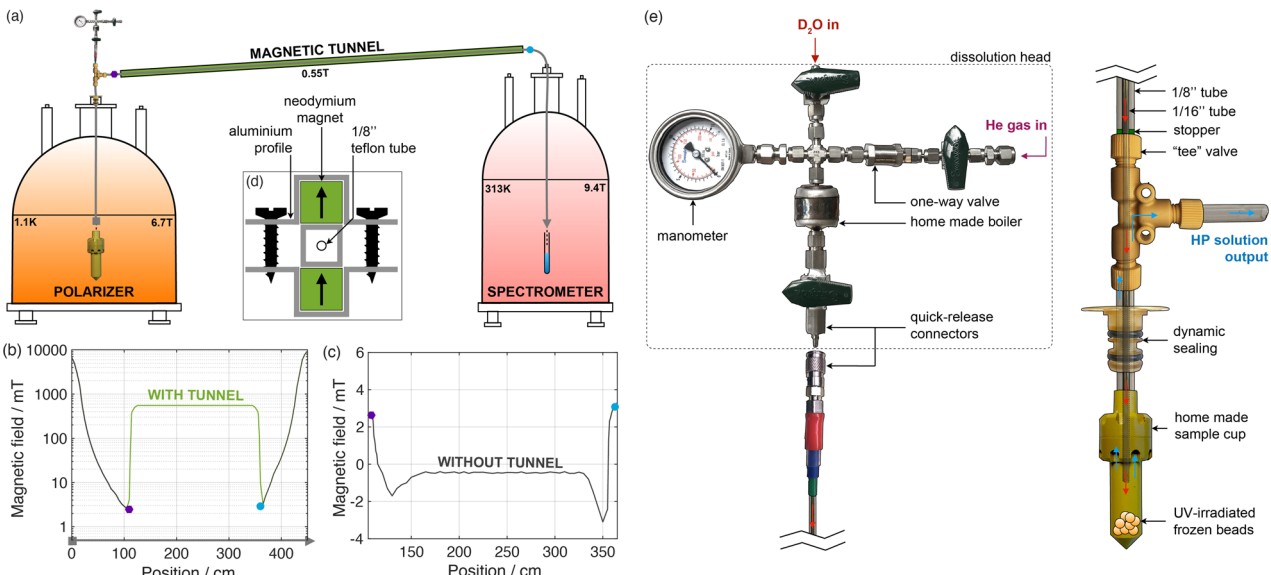

**Fig. 1 dDNP setup. a** Schematic representation of the dDNP set up composed of the 6.7 T polarizer, the magnetic tunnel, and the 9.4 T NMR magnet. The vertical component of the magnetic field measured along the sample pathway described by the gray arrow in panel **a** is reported with magnetic tunnel in panel **b** and without magnetic tunnel in panel **c**. In **a**, **b** and **c**, the purple hexagon and blue circle are markers to help guide the eye. **d** Schematic representation of the magnetic tunnel profile. **e** Custom fluid path (CFP) dissolution system.

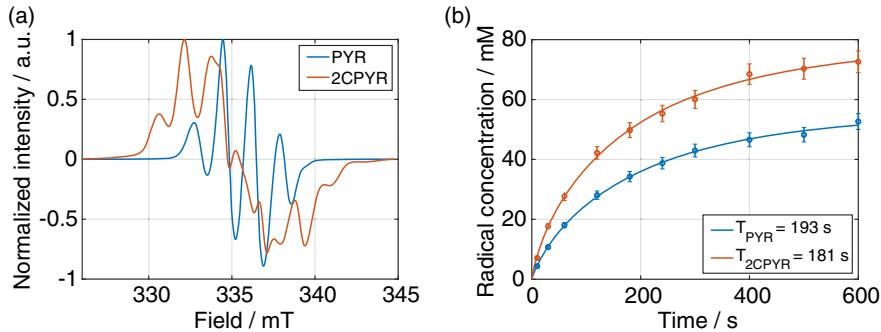

**Fig. 2 X-band ESR measurements. a** Normalized X-band ESR spectra after 10 min UV-light irradiation at 77 K and **b** radical generation time evolution of PYR_sample (blue circles) and 2CPYR_sample (orange circles). Data points and error bars are the average and standard deviation of repeated measurements from distinct samples ($n = 3$), respectively. The blue and orange curves were obtained by fitting the data to a mono-exponential function. The time constant resulting from the fits are reported in the inset. Error on fits was below 5%. X-band ESR spectra and radical generation time evolution of 2CPYRd_sample can be found in Supplementary Fig. 1.

on the solid-state DNP performance: [1]H concentration and UV-radical ESR linewidth. The results of the radical generation process are reported in Fig. 2. In panel **a**, the X-band ESR spectra of the PYR_sample and 2CPYR_sample are shown. The unpaired electron spin is largely localized on the PYR C2-carbon[33], and an additional hyperfine coupling of 5.6 mT was observed for 2CPYR, thus providing a broader spectrum compared to the PYR_sample. Panel **b** shows the radical generation time course. Frozen $6.0 \pm 0.5\,\mu L$ beads were irradiated in liquid nitrogen up to 600 s in batches of 8 beads ($48.0 \pm 4.0\,\mu L$ total sample volume). Radical yield and rate of formation depended on precursor concentration, UV-light power density and sample size[32]. To guarantee a final radical concentration of at least 50 mM, the radical precursor concentration was fixed to 20% of the final sample volume, and irradiation was performed using two deuterium UV-lamps of 19 W/cm$^2$ each. Although the radical generation build-up time constant was similar for PYR_sample and 2CPYR_sample (approx. 3 min), the latter consistently showed a 1.5 fold higher radical yield. Glassing of the DNP samples was achieved by adding 30% glycerol-$d_8$ and the pyruvic acid itself.

The 2PYRd_sample had a behavior very similar to its protonated counterpart (2PYR_sample). The only difference was that the additional protons present in the sample contributed to a slight extra broadening of the ESR spectrum. (Supplementary Fig. 1, panel **a**). All measurements were repeated at least 3 times. Detailed UV-sample preparation and handling was extensively reported previously[32], and summarized in Methods. Moreover, as comparison, a sample containing 50 mM of TEMPOL dissolved in glycerol-$d_8$:$H_2O$ 1:1 (v/v) was also prepared, from now onward referred as TEMPOL_sample. All numerical values are reported in Table 1.

**Solid-state DNP and LOD-ESR at 6.7 T and 1.15 K.** Fig. 3 shows the longitudinally detected (LOD) ESR spectrum (panel **a**) and [1]H DNP as a function of the microwave frequency (panel **b**) for PYR_sample, 2CPYR_sample and TEMPOL_sample. All measurements were performed at 6.7 T and $1.15 \pm 0.05$ K.

At these experimental conditions, the broadening of the ESR spectrum has two main contributions: anisotropy of the g-tensor,

Table 1 Sample properties, dDNP parameters, solid- and liquid-state results. Index "SS" corresponds to 6.7 T and 1.15 K, index "LS" corresponds to 9.4 T and 313 K. Columns 1 to 11: sample, water concentration in the SS, radical concentration, radical generation time constant at 77 K under UV-irradiation, LOD-ESR linewidth at 10% of the maximum measured at 6.7 T and 1.15 K, SS DNP ¹H steady-state polarization at 6.7 T and 1.15 K, LS DNP ¹H build-up time constant at 6.7 T and 1.15 K, LS ¹H polarization at 9.4 T and 313 K after dissolution and transfer, LS relaxation time constant at 9.4 T and 313 K, water concentration in the NMR tube, and the ¹H magnetization. Numerical values and errors are the average and standard deviation of repeated measurements from distinct samples (n = 3), respectively for column 3, 6, 8, 9, 10, and 11. Errors of other columns come from pipetting (column 2) and fitting precision (columns 4, 5, and 7). D indicates direct transfer without magnetic tunnel. M indicates manual transfer of a sample with high water concentration. A indicates the addition of sodium ascorbate to the solid-state sample. The "Water_sample" row corresponds to a reference sample of pure water at 9.4 T and 313 K.

| Sample | $[H_2O]_{SS}$ / M | [rad] / mM | $T_B$ / s | $\Delta_{ESR}$ / MHz | $P_{SS}$ / % | $T_{B,SS}$ / s | $P_{LS}$ / % | $T_{1,LS}$ / s | $[H_2O]_{LS}$ / mM | $M_{LS}$ / A.m⁻¹ |
|---|---|---|---|---|---|---|---|---|---|---|
| PYR_sample | 28 ± 0.5 | 52 ± 3 | 193 ± 31 | 238 ± 2 | 68.0 ± 3 | 881 ± 29 | 52.3 ± 2 | 33 ± 3 | 503 ± 112 | 4.5 ± 0.2 |
| 2CPYR_sample | 28 ± 0.5 | 73 ± 4 | 181 ± 23 | 391 ± 2 | 82.2 ± 3 | 377 ± 15 | 65.7 ± 5 | 31 ± 3 | 524 ± 101 | 5.8 ± 0.5 |
|  |  |  |  |  |  |  | 64.6 ± 5 D | 30 ± 2 D | 517 ± 83 D | 5.7 ± 0.4 D |
|  |  |  |  |  |  |  | 49.0 ± 8 M | 19 ± 1 M | 3117 ± 84 M | 26.0 ± 4.2 M |
| 2CPYRd_sample | 5.5 ± 0.1 | 74 ± 4 | 163 ± 30 | — | 96.9 ± 3 | 194 ± 7 | 75.5 ± 5 | 39 ± 2 | 196 ± 85 | 2.5 ± 0.2 |
| TEMPOL_sample | 28 ± 0.5 | 50 ± 0.5 | — | 459 ± 2 | 81.8 ± 3 | 309 ± 6 | 20.8 ± 4 | 8.4 ± 1 | 454 ± 79 | 1.6 ± 0.3 |
|  |  |  |  |  |  |  | 10.1 ± 3 D | 9.0 ± 1 D | 467 ± 66 D | 0.8 ± 0.2 D |
|  |  |  |  |  |  |  | 19.7 ± 3 A | 12.6 ± 1 A | 423 ± 62 A | 1.5 ± 0.2 A |
| Water_sample | — | — | — | — | — | — | 3.0 × 10⁻³ | 4.2 ± 0.5 | 55 ± 342 | 2.8 × 10⁻² |

and hyperfine coupling-tensor[25]. Results for the PYR_sample and 2CPYR_sample nicely reflected X-band measurements: the two UV-induced radicals were characterized by the same g-tensor[33], and the larger linewidth for the 2CPYR_sample (153 MHz broader than PYR_sample, values measured at 10% of maximum intensity, see Table 1) came from the extra hyperfine coupling to the ¹³C labeled C2-carbon. The TEMPOL_sample showed the broadest ESR spectrum (459 MHz) because of a larger g-tensor anisotropy (see Supplementary Fig. 3).

Given the temperature and radical concentrations, thermal mixing is expected to be the dominant DNP mechanism[38]. Indeed, for all samples, the DNP microwaves sweep reported in Fig. 3 reflected well the LOD-ESR spectrum: no DNP enhancement was observed at the center of gravity and beyond the extrema of the LOD-ESR spectrum. Moreover, since the electron $T_1$ ($T_{1e}$) of the three samples is relatively short (100–200 ms, Fig. 4, panel **d**), modulation of the microwave frequency promoted efficient spectral diffusion, improving the polarization enhancement (see Fig. 4, panel **a** to **c**)[39].

All samples were polarized at optimal microwave irradiation conditions in order to achieve the highest ¹H DNP enhancement: 188.20 GHz for PYR_sample, 187.92 GHz for 2CPYR_sample and 188.08 GHz for TEMPOL_sample at 55 mW output power. The microwave frequency was modulated at a rate of 1 kHz and an amplitude of 25 MHz for the UV-samples and 50 MHz for the TEMPOL_sample. Characteristic polarization build-up curves are reported in Fig. 5 panel **a**. While the 2CPYR_sample and TEMPOL_sample had a similar behavior reaching a solid-state proton polarization of 82 ± 3% and with a build-up time constant of 377 ± 15 s and 309 ± 6 s respectively, the PYR_sample reached a polarization value of 68 ± 3% with an almost three times longer build-up time constant.

The influence of the water proton concentration in the sample is illustrated in Fig. 4, panel **c**. The DNP curves of 2CPYR_sample and 2CPYRd_sample are reported. The sample formulations were the same except for the $H_2O$ content. In the 2CPYRd_sample, 80% of the water was replaced by $D_2O$ (proton concentration reduced from 56 M to 11 M), which halved the build-up time constant and allowed us to achieve 96.9 ± 3% proton polarization in the solid-state. All measurements were repeated three times and are reported in Table 1.

**Dissolution transfer and liquid-state relaxation**. Before dissolution, the samples were polarized by DNP until at least 95% of the plateau was reached (i.e., for three time constants of the exponential polarization build-up curve). 8 mL of $D_2O$ containing 0.1 g/L of Ethylenediaminetetraacetic acid (EDTA) was loaded into the dissolution head (Fig. 1, panel **e**), pressurized to 4 bars and then heated to approx. 190 °C (12 bars of vapor pressure). If paramagnetic ions were present in $D_2O$ due to the metallic structure of the boiler, EDTA would chelate these ions and inhibit their contribution to nuclear spin relaxation. While keeping the DNP polarizer sample space at approx. 1 mbar, the CFP was lifted 15 cm through the dynamic seal out of the liquid helium and connected to an exit tube traversing the magnetic tunnel from the polarizer to the 9.4 T NMR magnet. The CFP inlet was then connected to the dissolution head, the hot buffer released, and the HP solution flushed out of the polarizer under a constant pressure of 12 bars (DT transfer). The HP solution was eventually transferred directly into a 10 mm NMR. The latter was filled with 3.5 ± 0.1 mL of HP solution in approx. 2 s after releasing the hot buffer. Prior to dissolution and transfer, all tubing was carefully flushed with helium gas to eliminate $O_2$ and the 10 mm NMR probe set to 40 °C. Figure 5b shows the results of dissolution and DT transfer for the PYR_sample, 2CPYR_sample and TEMPOL_sample.

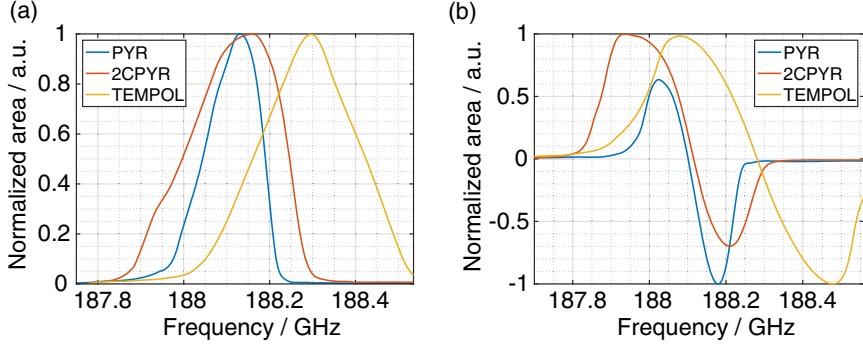

**Fig. 3 Solid-state LOD-ESR and DNP measurements. a** LOD-ESR spectrum and **b** $^1$H DNP microwave sweep spectra measured at 6.7 T and 1.15 K without microwave modulation are reported for PYR_sample (blue), 2CPYR_sample (orange), and TEMPOL_sample (yellow). The zero crossings in **b** have been corrected to coincide with the center of gravity (first moment of ESR spectrum) in **a**.

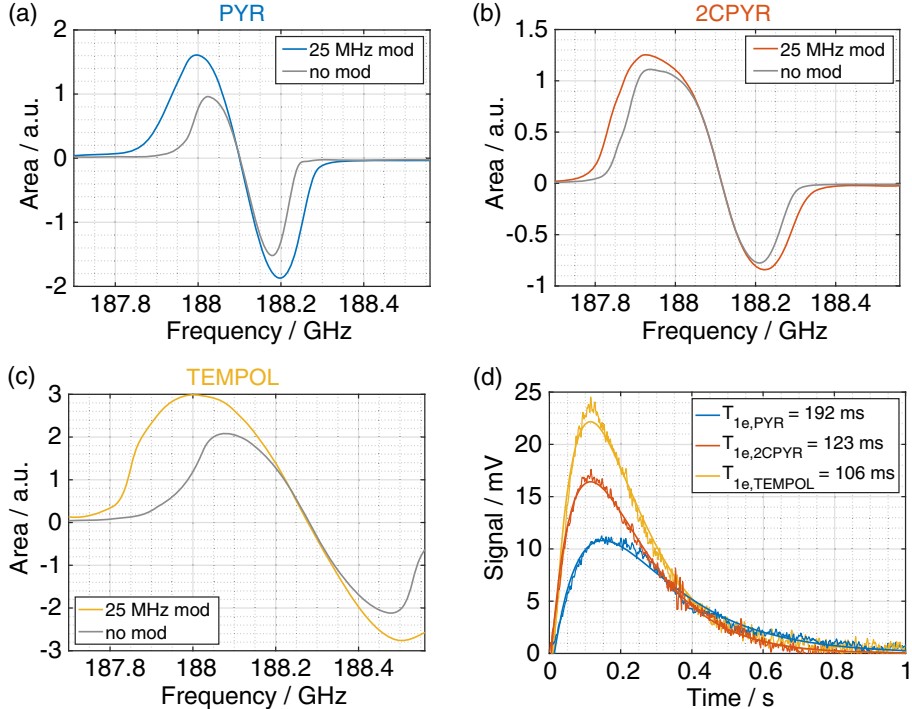

**Fig. 4 $^1$H DNP sweep spectra and $T_{1e}$ measurements.** $^1$H DNP microwaves sweep spectra measured at 6.7 T and 1.15 K with (color) and without (gray) microwave frequency modulation for PYR_sample (**a**), 2CPYR_sample (**b**), and TEMPOL_sample (**c**). **d** $T_{1e}$ measurements using the LOD-ESR probe for PYR_sample (blue), 2CPYR_sample (orange), and TEMPOL_sample (yellow). The experimental data was fitted (smooth curves) to the expression $S = A(\exp(-t/T_{1e}) - \exp(-t/\tau))$, where $\tau$ represents the pickup coil time constant and $A$ a proportionality factor[32]. Error on fit was below 5%.

The two UV-samples, with a measured liquid-state water proton polarization of $52 \pm 2\%$ and $66 \pm 5\%$ respectively, incurred a relative polarization loss of approx. 20%, while the TEMPOL_-sample lost more than 75% of its solid-state value. The 3 samples provided a final water concentration inside the 10 mm NMR tube of $493 \pm 79$ mM, but the liquid-state $T_1$ was only $9 \pm 1$ s for the TEMPOL_sample and $30 \pm 2$ s for the UV samples confirming the recombination of the UV-induced radicals into diamagnetic species during the dissolution process. Liquid-state $^1$H NMR spectra are shown in Supplementary Fig. 4.

As previously mentioned, sodium ascorbate can be employed to scavenge nitroxide radicals during the dissolution or at an intermediate stage between dissolution and injection of the HP solution[13,40,41]. Although the absence of radical in the final solution can increase the HP water $T_1$, the relatively slow kinetic of the reaction between the radical and the scavenger does not alleviate from severe polarization losses[13]. To verify this, we performed a control experiment following methods described previously[41]. The TEMPOL_sample was transferred to the CFP sample cup together with an identical volume of a frozen solution of 1.5 M sodium ascorbate in $D_2O$ in order to obtain, after dissolution and mixing of the two parts, a 1:30 radical-ascorbate ratio[41]. As shown in Supplementary Fig. 3 panel f, the water $T_1$ increased from $8.4 \pm 1$ s to $12.6 \pm 1$ s, but the measured polarization was similar (i.e., $19.7 \pm 3\%$). The reason for the water $T_1$ being shorter than the dissolved UV-samples, can be ascribed to the presence of an additional relaxation pathway due to the presence of 10.5 M ascorbate protons in the final solution.

To verify that paramagnetic relaxation, especially at low field, was the main source of polarization loss, we repeated the experiments removing the magnetic tunnel (D transfer). While for the PYR_sample and 2CPYR_sample there was essentially no difference between a DT transfer and a D transfer, for the TEMPOL_sample the liquid state polarization was further

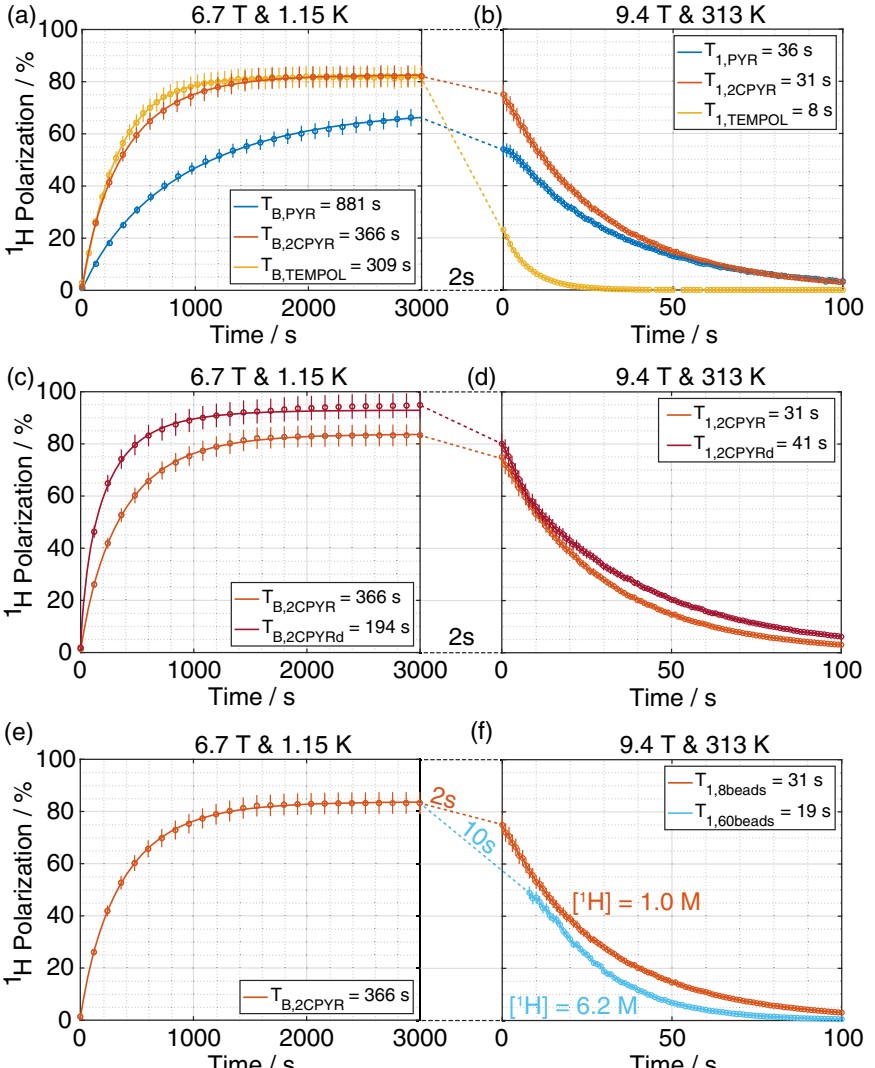

**Fig. 5 Solid-state DNP build-up and liquid-state relaxation measurements.** Solid-state $^1H$ polarization build-up comparison at 6.7 T and 1.15 K between PYR_sample (blue circles), 2CPYR_sample (orange circles) and TEMPOL_sample (yellow circles) (**a**) and between 2CPYR_sample (orange circles) and 2CPYRd_sample (red circles) (**c**). Liquid-state relaxation comparison at 9.4 T and 313 K after DT transfer between PYR_sample (blue circles), 2CPYR_sample (orange circles) and TEMPOL_sample (yellow circles) (**b**) and between 2CPYR_sample (orange circles) and 2CPYRd_sample (red circles) (**d**). Panel **e** and **f** show the effect of increasing $^1H$ nuclei concentration in the liquid state. Data points and error bars are the average and standard deviation of repeated measurements from distinct samples ($n = 3$), respectively. All curves were obtained by fitting the data to a mono-exponential function. The different time constants resulting from the fits are reported in the insets. Error on fits was below 5%.

reduced to half, leaving a liquid-state water polarization of $10 \pm 3\%$ only (Supplementary Fig. 3). This was in good agreement with previous studies showing water paramagnetic relaxation to be very effective below 0.1 T[42,43]. Fig. 5 panel **d** shows the comparison between 2CPYR_sample and 2CPYRd_sample. Dissolution and D transfer of the latter generated a polarization loss comparable with the other UV-samples, but the preserved $75 \pm 5\%$ liquid-state proton polarization (corresponding to a proton enhancement of ca. 25,000) relaxed with a longer $T_1$ of $39 \pm 2$ s due to lower proton concentration ($196 \pm 85$ mM) and thus reduced $^1H$ homonuclear dipolar relaxation.

We investigated increasing the water concentration and relieving the transfer time. The sample cup was filled with 60 frozen beads (i.e., 360 µL instead of 48 µL) of 2CPYR_sample. The HP solution was collected next to the polarizer and manually injected into the 10 mm NMR tube. The latter was then inserted into the 9.4 T NMR magnet and the NMR acquisition started approx. 10 s after release of the hot buffer inside the CFP. A $49 \pm$

$8\%$ liquid-state polarization with a $T_1$ of $19 \pm 1$ s was measured for a final water concentration of $3.12 \pm 0.08$ M (Fig. 5, panel **f**). This sample showed the highest magnetization obtained in this study ($M_{LS} = 30.2$ A.m$^{-1}$ at best, see Table 1, details in Supplementary Information). This represents a magnetization enhancement of 1070 compared to a sample of pure water measured at identical conditions. Therefore, it would require a magnetic field of $1070 \cdot 9.4 = 10{,}062$ T to achieve such a water signal with a sample of pure water at room temperature. All measurements were repeated at least three times and are reported in Table 1.

To demonstrate the potential of our new methodology relying on non-persistent radicals we repeated an experiment application employing HP water, previously performed by Harris et al.[17], using nitroxide radicals as polarizing agent. HP water is injected into a NMR tube containing a solution of urea dissolved in $D_2O$; the high proton magnetization is spontaneously transferred to the exchangeable protons of urea that in turn enhance the

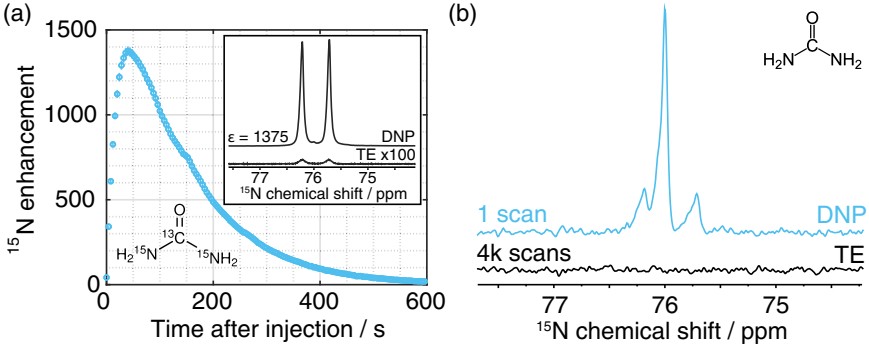

**Fig. 6 Polarization of heteronuclei measurements. a** $^{15}$N enhancement as a function of time after injection of hyperpolarized water sample (60 beads of 2CPYR_sample, D transfer) into a 10 mm NMR tube containing 500 µL of 400 mM [$^{13}$C,$^{15}$N$_2$]urea, acquired at 9.4 T and 313 K. The inset shows the comparison between the DNP and thermal equilibrium $^{15}$N spectra. **b** DNP enhanced (SNR = 80) and thermal equilibrium natural abundance $^{15}$N spectra of 500 µL of 400 mM urea acquired with a single 90° scan 40 s after the injection of the hyperpolarized water sample (60 beads of 2CPYR_sample, D transfer). In **a** and **b**, the $^{15}$N resonances at 76.2 and 75.7 ppm are due to the $^{15}$N-$^{13}$C coupling (20 Hz coupling).

polarization of directly bonded $^{15}$N nuclei via heteronuclear cross relaxation (i.e., Nuclear Overhauser Effect). Thus, the polarization from DNP-enhanced water flows spontaneously to $^{15}$N nuclei with no need for any $^{1}$H pulsing[17]. 60 beads of the 2CPYR_sample was polarized, dissolved and transferred, without magnetic tunnel, directly into a 10 mm NMR tube containing 500 µL of 400 mM [$^{13}$C,$^{15}$N$_2$]urea in D$_2$O. The urea concentration was comparable to Harris et al. Fig. 6a shows the time course of $^{1}$H–$^{15}$N polarization transfer. The high proton magnetization and long relaxation times led to a maximum $^{15}$N enhancement of 1375±28 after approx. 40 s, i.e., 3.4 times higher than previously reported[17]. Moreover, the experiment was repeated at identical concentrations, but employing natural abundance urea. Results are reported in Fig. 6b. In this latter case a single π/2-pulse, 40 s after the injection of the HP water sample was used to record the $^{15}$N signal. A signal-to-noise-ratio of 80 was obtained for this low-gamma dilute spin system ([$^{15}$N] ≈ 366 µM). The spectrum shows singlet from $^{15}$N-urea at natural abundance (2 × 0.36%) and the doublet from the $^{13}$C,$^{15}$N-urea (2 × 0.36% × 1%).

## Discussion

We disclose hyperpolarization of water protons using labile UV-induced radicals and dDNP. We investigate and report on critical parameters necessary to optimize the method and understand the physics.

The results show that solid-state DNP and build-up time constants for the UV-induced radicals are comparable to what has been achieved using the best stable, polarizing agents for protons, i.e., nitroxides. The solid-state polarization, using 2CPYR as radical precursor has the advantage of working with a radical species characterized by a broader ESR spectrum than PYR. In thermal mixing the polarization to nuclei is transferred from electron spin pairs with opposite orientation and separated by one nuclear Larmor frequency. Therefore, DNP efficiency depends on the autocorrelation integral of the radical ESR spectrum evaluated at the $^{1}$H Larmor frequency ($\int_{-\infty}^{\infty} g(\omega) \cdot g(\omega - \omega_{1H}) d\omega$, where $g(\omega)$ is the radical ESR spectrum function). The latter provides an estimation of the number of effective electron spin pairs: for a given radical concentration, the higher is their number the more effective the polarization transfer to the nuclei. The ESR spectrum autocorrelation integral of the 2CPYR_sample was comparable to the TEMPOL_sample. On the other hand, for the PYR_sample, the integral was one order of magnitude lower than the 2CPYR_sample (see Supplementary Information), as suggested

by the LOD-ESR width of 238 ± 2 MHz compared to the $^{1}$H Larmor frequency of 285.5 MHz at 6.7 T. Moreover, the radical yield in the 2CPYR_sample was also higher. Although we do not find an explanation for this effect other than the higher purity of the $^{13}$C-labeled compound compared to natural abundance one, the consequence was twofold. First, the larger amount of radical compared to the TEMPOL_sample compensated for the narrower ESR spectrum achieving very similar solid-state DNP performance. Second, the increased dipolar coupling between electron spins made microwave modulation less effective in achieving high polarization.

The 2CPYRd_sample reached the highest polarization in the shortest time. Protons are notoriously "heavy" nuclei to polarize via thermal mixing and partial or full deuteration of the sample can help in achieving higher nuclear polarization when broad ESR line radicals are involved[21]. Indeed, the polarization build-up time is, within alike spins, proportional to $N_I$, where $N_I$ is the nuclear spin concentration[25]. Keeping all other parameters unchanged, decreasing the proton content in the sample speeded up the polarization transfer making the DNP mechanism more efficient[21,44].

Concerning the dissolution and transfer of the HP solution, we would like to stress the irrelevance of the magnetic tunnel when employing UV-samples, and thus the more straightforward implementation of HP water in any MR facility. The advantage of a polarizing agent that immediately recombines into diamagnetic species as soon as the hot buffer gets in contact with the frozen sample, not only improved the water proton polarization by a factor of 6 (up to 75.5 ± 5 % $^{1}$H polarization) over state-of-the-art, but also demonstrated that a low field magnetic environment during transfer of the HP solution is not a limiting factor for preserving the enhancement.

We would like to emphasize that from an application perspective, what matters is the proton magnetization available at time of use. The final water concentration was calculated from the proton H$_2$O NMR linewidth by means of a calibration curve obtained from a series of D$_2$O:H$_2$O solutions with known ratio (0.1 to 56 M). Therefore, although the best polarization was obtained for 8 beads of 2CPYRd_sample, we employed the sample with the highest magnetization in our application: 60 beads of 2CYPR_sample. Herein, we demonstrated that the high magnetization preserved after dissolution (26 A/m) could enhance the $^{15}$N NMR signal of urea by more than 1300-fold after proton exchange with HP water. The latter represents an improvement by a factor 3.4 compared to state of the art[17]. Moreover, the same experiment was performed on a dilute spin system ([$^{15}$N] = 366 µM) employing natural abundance urea as

target substrate. In this case, the water magnetization was high enough to acquire a $^{15}N$ spectrum with a SNR of 80 in a single scan.

Our study clearly shows that the combination of broad ESR line UV-induced labile radicals with dDNP represents the best method for hyperpolarizing water to date. The unprecedent liquid-state nuclear spin magnetization obtained with no need for any radical filtration or "magnetic sheltering" of the sample pushes hyperpolarization of water to the next level. The number of applications employing HP water already claims a continuously growing track record[45]. The clear improvement and ease of operation we introduced may open up for new frontiers in medicine, biology and chemistry.

## Methods

**UV-samples preparation**. All chemicals were purchased from Sigma-Aldrich (Denmark). UV-samples mixtures were prepared in an Eppendorf tube, homogeneously mixed, and then sonicated at 40 °C for 5 min to efficiently degas the solutions to improve the glass quality after freezing. Immediately after, a volume of 6.0 ± 0.5 μL was taken from the Eppendorf tube by means of a micropipette (1–20 μL) and added as a drop to liquid nitrogen to form one frozen bead. The operation was repeated until the desired number of beads was obtained. Beads were transferred in batches of 8 inside a synthetic quartz Dewar (Miniscope MS 5000 ESR spectrometer compatible, Magnettech, Berlin, Germany) filled with liquid nitrogen for UV irradiation. The UV-irradiation setup was extensively described previously[32]. The only difference in this study was the opening of a second irradiation port to use two 19 W/cm$^2$ broad band deuterium lamps at the same time (Dymax BlueWave 75, Torrington, CT USA). The UV sources were always operated at full power to provide the highest achievable radical yield. Samples were irradiated until a radical concentration plateau was attained (approx. 600 s).

**ESR experiment and radical quantification**. For all the experiments, the X-band ESR MiniScope 5000 spectrometer (Magnettech, Berlin, Germany) was used. The spectrometer parameters, kept constant for all measurements, were optimized to avoid any saturation or line broadening of the ESR signal, i.e., center of the sweep = 338 mT; sweep range = 20 mT; sweep time = 20 s; modulation frequency = 100 kHz; modulation amplitude = 0.1 mT; and microwave power = 0.2 mW. The radical concentration was calculated from the spectrum double integral by means of a calibration curve obtained from a series of 6.0 ± 0.5 μL frozen beads of glycerol-$d_8$: $H_2O$ (5:5) with known concentrations of 4-hydroxyTEMPO (12.5−100 mM). All measurements were repeated three times. Data were processed in MATLAB (Mathworks, Natick, MA, USA).

**UV-samples handling and loading into the polarizer**. After UV-irradiation, samples were poured into a semi-spherical glass dewar filled with liquid nitrogen. From there the beads were transferred inside the CFP homemade vial (see Fig. 1, panel e). The vial is divided into two parts made of Polyamide-imide (PAI) plastic: the neck and the body, able to contain up to 0.5 mL of sample. The vial neck was attached to the external surface of the Polyether ether ketone (PEEK) outer lumen (ID: 0.072 ± 0.002 in, OD: 0.125 ± 0.002 in) by means of the UV-adhesive Dymax 215-C (Dymax, Torrington, CT, USA); the latter was cured for 30 s using the same UV lamp employed to generate the radicals. A 3D-printed wrench immersed halfway in liquid nitrogen was used as a stand for the vial body during the transfer of the frozen beads. A PTFE O-ring was then placed and compressed between the neck and the body by screwing the first into the second to form a helium leak-tight closure. The fluid path was then flushed with helium gas, checked for leak tightness, and inserted into the polarizer. Upon reuse, the vial was opened and dried, and only the PTFE O-ring was replaced with a new one.

**Solid-State DNP**. All DNP measurements were performed on a home-built dDNP polarizer operating at 1.15 ± 0.05 K and 6.7 T (Magnet and cryostat from Magnex Scientific Ltd, Yarnton, UK). Microwaves were delivered from a 94 GHz solid-state source VCOM-10/94-WPT (ELVA-1, St. Petersburg, Russia) coupled to a 200×2R4 frequency doubler (VDI, Charlottesville, VA, USA), which provided an output power of 55 mW at 188 GHz. The source, digitally controlled through NI-DAQ device USB-6525 (National Instruments, Austin, TX, USA) had a tuning range of ±0.6 GHz and the possibility to modulate the output frequency at a rate up to 2 kHz and with an amplitude of up to 100 MHz. All $^1H$ NMR acquisitions were performed using a Varian INOVA console (Palo Alto, CA, USA) connected to a low-temperature probe modified with respect to the original version[7] to accommodate a Custom Fluid Path (CFP). The flip angle used for all acquisitions was 1° (pulse length = 5 μs; transmitted power = 5 W). The microwave frequency giving the maximum DNP enhancement was found by sweeping the latter in steps of 5 MHz. At each microwave frequency, the build-up lasted for 30 min; afterwards, the NMR signal was destroyed with a comb of 50,000 rf pulses separated by 40 μs before passing to the next frequency step. The polarization build-up was monitored

by pulsing every 60 s or 120 s. After having switched off the microwaves and saturated any residual signal with the 50,000 rf pulses comb, the thermal equilibrium build-up was monitored overnight. The NMR signal was acquired every 30 min (1 average) until complete relaxation was achieved. The DNP enhancement was calculated by dividing the thermal equilibrium and DNP signal integrals. All of the measurements were repeated at least three times. All of the data were processed with MNOVA (Mestrelab Research, Santiago de Compostela, Spain) and MATLAB (Mathworks, Natick, MA, USA).

**LOD-ESR Measurement**. ESR spectra of UV-irradiated samples were measured at real DNP conditions (i.e., 6.7 T and 1.15 ± 0.05 K) using a homemade setup for longitudinal detection (LOD) described previously[32]. The ESR spectrum was obtained by sweeping the microwave frequency over the full range in steps of 5 MHz. For each frequency, the output power was square wave modulated from 0 to 55 mW at a frequency of 5 Hz. The intensity of the demodulated signal, proportional to the number of electron spins resonating at the given frequency, was plotted as a function of the microwave frequency. A frequency of 5 Hz was found to be a good compromise between the efficiency of the lock-in amplifier and the intensity of the signal. Modulating at higher frequency, beneficial from the lock-in point of view, caused the saturation of the electron spins and a reduction of the signal. On the other hand, a slower modulation (0.5 Hz) of the microwave power allowed visualization of the full signal evolution across the detection coil induced by the electron spins. This procedure was used to measure the electron spins $T_{1e}$ at a given microwave frequency (see Fig. 5). The induced voltage time evolution was fitted (smooth curves) to the expression $S = A(\exp(-t/T_{1e}) - \exp(-t/\tau))$, where $\tau$ represents the pickup coil time constant and $A$ a proportionality factor.

**Magnetic tunnel**. The magnetic tunnel was build using 160 permanent $30 \times 12 \times 12$ mm$^3$ magnets from Supermagnete (Gottmadingen, Germany). The magnets were forming two rails separated by a 10 mm aluminum squared profile. Both rails were magnetized in the same direction, providing a homogeneous 0.55 T magnetic field inside the aluminum profile (Fig. 1, panel **d**).

**Dissolution and liquid-state measurements**. Helium gas was slowly bubbled for 5 min inside the dissolution buffer—8 mL of a solution of $D_2O$ containing 0.1 g/L of Ethylenediaminetetraacetic acid (EDTA)—in order to remove most of the $O_2$. The solution was then loaded into the CFP dissolution head (Fig. 1, panel e) and pressurized to 4 bars with helium gas. The solution was heated to approx. 190 °C (12 bars of vapor pressure). While keeping the DNP polarizer sample space at approx. 1 mbar, the CFP was lifted 15 cm through the dynamic seal out of the liquid helium and connected to the exit tube. The CFP inlet was then connected to the dissolution head, the hot buffer released, and the HP solution flushed out of the polarizer under a constant pressure of 4 bars (DT and D transfer). The HP solution was directly transferred into a 10 mm NMR tube placed inside the 9.4 T NMR magnet with the exit tube placed inside the magnetic tunnel. The superheated buffer reached the sample slowing through the CFP inner lumen made of PEEK (ID: 0.062 ± 0.002 in, OD: 0.072 ± 0.002 in). The melted sample came out from the polarizer flowing in between CFP inner and outer lumens (Polyphenylsulfone ID: 0.095 in, OD: 0.125 ± 0.002 in). It finally reached the NMR tube placed inside the 9.4 T magnet flowing through a PTFE exit pipe (ID: 0.063 ± 0.002 in, OD: 0.125 ± 0.002 in) connected to the CFP outlet. The exit pipe and NMR tube were also flushed with helium gas for 1 min prior to dissolution to eliminate oxygen from the system. Even though 8 mL of buffer were loaded into the boiler, only 3.5 ± 0.1 mL filled the NMR tube due to the dead volume of the transfer line. The final water concentration was calculated from the proton $H_2O$ NMR linewidth by means of a calibration curve obtained from a series of $D_2O$:$H_2O$ solutions with known ratio (0.1 to 56 M).

## Data availability
Raw data are available upon request from the corresponding author.

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

## Acknowledgements

The authors thank Magnus Karlsson and Mathilde H. Lerche for helpful discussions and advice, as well as Jonas Milani and Jan Kilund for building the magnetic tunnel and technical support. This work received funding from the Danish National Research Foundation (DNRF124). This project has received funding from the European Union's Horizon 2020 research and innovation program under grant agreements No 713683 and 858149.

## Author contributions

A.C.P. and A.C. equally contributed to this study, they performed research, analyzed data and wrote the paper. J.H.A. designed the study and co-wrote the paper.

## Competing interests

The authors declare no competing interest.
