## [Peer Review File · Communications Chemistry]

This manuscript has been previously reviewed at another Nature Research journal. This document only contains reviewer comments and rebuttal letters for versions considered at Communications Chemistry.

REVIEWERS' COMMENTS:

Reviewer #1 (Remarks to the Author):

Overall, I am satisfied with the authors' responses, and believe that the manuscript is ready for publication.

That said, I still disagree with the statement in the title, now even more so once the authors have provided an explicit explanation of how the 10,000 T value was derived. They commented that providing NMR signal enhancement says nothing about concentration, which is correct. However, their version of the title is in fact misleading as far as concentrations are concerned. Specifically, at 10 T their 3M sample gives a 1000-fold stronger signal than pure water (55 M). This is where the 10000 T (= 10 T x 1000) value comes from. However, from the title one would most likely get an impression that for the actual sample used in the experiments, the field of 10,000 T would be required to get the same signal intensity at thermal equilibrium. This is incorrect; to get the same signal intensity for a 3M thermal sample, one would require an equivalent of $10,000 \text{ T} \cdot 55/3 = 180,000 \text{ T}$ (!). Sample magnetization is indeed important, but nowhere in the title is magnetization mentioned.

I hope the authors would reconsider, but at this point I wouldn't insist.

One minor correction – “enhancement of c.a. 25,000” should be changed to “enhancement of ca. 25,000”

Response to reviewer

- The title has been changed as requested by the reviewer.
- c.a. changed to ca.

Response to reviewers

Reviewers' comments:

Reviewer #1 (Remarks to the Author):

The authors have substantially improved this manuscript by addressing the minor issues raised in the previous review as well as several of the major criticisms. However, the primary criticism I and another reviewer raised—that this work represents incremental improvements in polarization rather than a new, novel breakthrough in either approach or application—remains. The inclusion of ^{15}N polarization at natural abundance in urea increases the impact of the paper, but I still don't think this result raises this work to the level of publication in Nature Comm. Again, without a true biological or biochemical application, it is difficult to argue that the impressive gains in water polarization seen in model systems will translate into an arena where they can truly transform magnetic resonance measurements. Also, in their point-by-point response to previous criticisms, the authors are somewhat inconsistent in how they respond. For example, they claim the addition of NaOH to pH balance a sample with TEMPOL would make it too different of a sample for direct comparison to pyruvate radical under the same glassing/deuteration/pyruvate conditions yet they also claim that the choice of glassing agent has little effect on attainable polarization. (Prior published literature demonstrates that glassing conditions can influence outcomes.)

Answer: The TEMPOL sample was included as a reference for best achievable solid- and liquid-state polarization with a nitroxide radical for hyperpolarized water, and not to be directly comparable to the UV sample.

The additional experiments added to compare controls do strengthen the rigor of the study yet at the same time underscore that the gains demonstrated by careful implementation of deuteration, pyruvate radical, deoxygenation, etc., etc. are only adding a few-fold gain in polarization and individually all these approaches have already been published. Ultimately, these additional gains, while helpful, are not going to majorly impact measurements in complex biological/biochemical systems where the primary drivers of proton relaxation will still exist. For these reasons, I recommend this work be submitted to a more specialized journal.

Reviewer #2 (Remarks to the Author):

1) The title - it is certainly not an easy task to choose an appropriate one. Still, I believe that the title chosen by the authors is not the best possible one. Reporting the achieved signal improvement as the field equivalent can be highly misleading, especially to the broad audience of non-NMR specialists.

Answer: We believe that the Tesla equivalent is the most correct and descriptive measure for achieved magnetization, which is the quantity of significance here. The meaning of “tesla equivalent” is explained in the abstract: “we report water signals otherwise requiring 10,000 T

at room temperature.“ We have further edited the title to indicate that the article subject is hyperpolarized magnetic resonance of water.

It is not obvious whether or not this value reflects the achieved concentration, especially provided that the authors never explain how they derived the 10000 T value.

Answer: We have thoroughly explained how the equivalent Tesla is calculated in the body text. The following sentences were added to the main text: “This sample showed the highest magnetization obtained in this study ($M_{LS} = 30.2 \text{ A.m}^{-1}$ at best, see Table 1, details in Supplementary Information). This represents a magnetization enhancement of 1,070 compared to a sample of pure water measured in identical conditions. Therefore, it would require $1,070 \cdot 9.4 = 10,062 \text{ T}$ to achieve such a water signal with a sample of pure water at room temperature.” We also added in the supplementary information a sentence explaining how the magnetization was calculated: “The magnetization was calculated as $M_{LS} = \frac{1}{2} \gamma h C N_A P$ where γ is the proton gyromagnetic ratio, h Planck’s constant, C the proton concentration, N_A Avogadro’s number, and P the liquid-state polarization.”

In my opinion, signal enhancement in combination with the measurement field and/or frequency would be the best choice. After all, the seminal dissolution DNP paper (certainly known to the authors) reported the >10000 SNR enhancement, even without reference to the nucleus and the measurement field.

Answer: Enhancement at a given field strength does not take into account the concentration of the nuclear spin. Our aim is maximizing magnetization.

In addition, please provide the achieved enhancement for ^1H at least once somewhere in the main text in the appropriate context, not only in the supplementary file. I believe that this is quite important, especially now that the ^{15}N enhancement is given in the paper.

Answer: The following was added: “(corresponding to a proton enhancement of c.a. 25,000)”

2) “Answer: We agree with the reviewer that a calculation using 8 mL of buffer would lead to a lower proton concentration. Nevertheless, during the dissolution process, the sample is mixed with the very first milliliter of D_2O ...”

This is a surprisingly naïve description of the entire dissolution and sample transfer process. Dissolution of a cryogenic solid in a very cold container by a few ml of overheated liquid is certainly a much more complex process. Furthermore, during the fast sample transfer, the flow of the liquid in the transfer channel is likely highly turbulent. Therefore, the provided calculation of the final concentration can be off by a large margin. In my experience, the final concentration of the solute can be (significantly) different (in fact, lower) than a simple estimate based on the total volume of the dissolution medium (or even the smaller volume that arrives to the NMR tube) would suggest. As far as I understand, the final concentration of H_2O (HDO , H) was actually measured, which is a lot more reliable, and maybe it is worth stressing at least once in the discussions that this is a measured value.

Answer: We agree with the reviewer and have amended the text. The concentration is measured. The dissolution process is complicated (heterogeneous), but the final volume is homogeneous. Since the complete solid sample is dissolved in the recovered volume, the calculated concentration agrees well with the measured. We added the following sentence in

the discussion section: “The final water concentration was calculated from the proton H₂O NMR linewidth by means of a calibration curve obtained from a series of D₂O:H₂O solutions with known ratio (0.1 to 56 M).”

The fact that it is similar to the one yielded by the simplistic calculation is immaterial in my opinion. Also, I’d expect that evaluation of the water proton concentration would be more reliable if performed by integrating the NMR signal and comparing it to a reference sample, not measuring the signal width. Any particular reason why this was not done by signal integration?

Answer: We have found the line width method to be robust when the line width changes from kHz to Hz due to radiation damping (choice of integration region). The two methods provide the same result.

3) In the sentence “The samples were polarized by DNP until at least 95% of the signal plateau was reached (i.e. for three time constants of the exponential curve) before dissolution”, I suggest to change “exponential curve” to “exponential polarization build-up curve”

Answer: The sentence was modified

4) “Indeed, the polarization build-up time is proportional to $N_I \times (\omega_{I})^2$.”
I believe, this is not the entire story, and in particular the relaxation times should matter as well. Otherwise, ¹³C polarization buildup would be a lot faster than that for ¹H. However, cross-polarization from ¹H to ¹³C nuclei is often used to polarize ¹³C nuclei faster compared to direct ¹³C polarization by taking advantage of faster ¹H polarization and an “instantaneous” ¹H to ¹³C polarization transfer by a CP contact.

Answer: We meant that, within alike spins, the polarization process is faster when the number of spins to polarize decreases. We changed the sentence: “Indeed, the polarization build-up time is, within alike spins, proportional to N_I , where N_I is the nuclear spin concentration.”

5) “and allowed us to achieve 96.9±3 % proton polarization.” Please, add “in the solid state” to avoid even a slightest possibility of confusion.

Answer: The sentence was modified

6) Line 225: “(Supplementary Figure 5, panel f).”
There is no such figure. The reference is likely made to Figure 5 of the main text.

Answer: The sentence was corrected to “Figure 5, panel f”

7) “In the 2CPYRd_sample, 80% of the water was replaced by D₂O (H₂O concentration reduced from 56 M to 11 M).”
This is incorrect. The concentration of H₂O (not protons) in the 2CPYR_sample is 28 M, not 56 (50% H₂O by volume), therefore in 2CPYRd_sample it amounts to ca. 5.5 M. Please, check all numbers in the manuscript for consistency.

Answer: The sentence was corrected: “(proton concentration reduced from 56 M to 11 M)” and the numbers were checked.

8) Please, spell parahydrogen properly, as a single word, without a hyphen. This is the only correct way of spelling it.

Answer: The word was corrected.

A couple of less important issues are:

9) “the longitudinal detected (LOD) ESR” should be “the longitudinally detected (LOD) ESR”

Answer: The sentence was corrected.

10) “The factor 3.6 improvement from 20.8 % to 75.5 % (Table 1) would lead to a very significant improvement in image quality in an MRI experiment.”
Once again, “very significant” is a matter of perception that can be looked at from a different perspective. A 1000-fold NMR signal enhancement would certainly be very significant as it would provide a 10-fold isotropic improvement in spatial resolution of an MR image. At the same time, a 3.6-fold signal enhancement would provide a ca. 1.5-fold resolution improvement in MRI, which is a lot less significant. Alternatively, one could sacrifice a factor of 1.5 in spatial resolution to get the same SNR improvement in an image without the need to resort to the sophisticated hyperpolarization technique. Therefore, if anything, in my opinion a factor of 3.6 signal enhancement is more significant for spectroscopic rather than imaging experiments.

Reviewer #3 (Remarks to the Author):

The revised version of the manuscript addresses the concerns raised by the reviewers adequately, including my own. I recommend publication in the present form, with one exception: The title. While I am still sceptical about the merit of quantifying the achievement in terms of an “equivalent magnetic field”, I do have to admit that it is difficult to find alternatives that are both correct and informative. However, I am still concerned that the title gives very little indication about the nature and context of the work done. This is particularly problematic in a journal with a broad audience. The reader is left to infer from the mention of "Tesla" and "proton" that the article is about magnetic resonance. The initiated may be able to make that conclusion, but it will not come naturally to the vast majority of readers of Nature Communications, who have different scientific backgrounds. How about something much more descriptive, like "Dramatic Signal Enhancements of Water in Magnetic Resonance Imaging by Dissolution Dynamic Nuclear Polarisation"? The argument about the 10000 Tesla "equivalent" magnetic field may still be made in the abstract, introduction, and/or conclusions, but now with the appropriate caveats and qualifications to avoid misleading non-experts.

Answer: We have further edited the title to indicate that the article subject is hyperpolarized magnetic resonance of water.